# Pulmonary Manifestations of Plasma Cell Type Idiopathic Multicentric Castleman Disease: A Clinicopathological Study in Comparison with IgG4-Related Disease

**DOI:** 10.3390/jpm10040269

**Published:** 2020-12-10

**Authors:** Midori Filiz Nishimura, Takuro Igawa, Yuka Gion, Sakura Tomita, Dai Inoue, Akira Izumozaki, Yoshifumi Ubara, Yoshito Nishimura, Tadashi Yoshino, Yasuharu Sato

**Affiliations:** 1Department of Pathology, Okayama University Graduate School of Medicine, Dentistry, and Pharmaceutical Sciences, Okayama 700-8558, Japan; p2hq21br@s.okayama-u.ac.jp (M.F.N.); me19075@s.okayama-u.ac.jp (T.I.); yoshino@md.okayama-u.ac.jp (T.Y.); 2Division of Pathophysiology, Okayama University Graduate School of Health Sciences, Okayama 700-8558, Japan; gion@okayama-u.ac.jp; 3Department of Pathology, Tokai University School of Medicine, Kanagawa 259-1193, Japan; hs800759@tsc.u-tokai.ac.jp; 4Department of Radiology, Kanazawa University Graduate School of Medical Sciences, Ishikawa 920-8641, Japan; d-inoue@lake.ocn.ne.jp (D.I.); aizumozaki@gmail.com (A.I.); 5Nephrology Center, Toranomon Hospital Kajigaya, Kanagawa 213-0015, Japan; ubara@toranomon.gr.jp; 6Department of General Medicine, Okayama University Graduate School of Medicine, Dentistry, and Pharmaceutical Sciences, Okayama 700-8558, Japan; nishimura-yoshito@okayama-u.ac.jp

**Keywords:** plasma cell type idiopathic multicentric Castleman disease, IL-6, IgG4-related disease, immunohistochemistry, hyper IL-6 syndrome

## Abstract

Plasma cell type idiopathic multicentric Castleman disease (PC-iMCD) occasionally manifests as parenchymal lung disease. This study aimed to elucidate the detailed clinicopathological features of lung lesions in PC-iMCD and compare the findings with those in immunoglobulin (Ig) G4-related disease (IgG4-RD), the most difficult differential diagnosis of PC-iMCD. We analyzed the clinicopathological findings and immunohistochemical expression patterns of interleukin-6 (IL-6) and Igs in lung specimens from 16 patients with PC-iMCD and 7 patients with IgG4-RD. Histologically, pulmonary PC-iMCD could not be differentiated from IgG4-RD based on lesion distribution patterns, the number of lymphoid follicles and obliterative vasculitis, or fibrosis types. The eosinophil count was higher in the IgG4-RD group than in the PC-iMCD group (*p* = 0.004). The IgG4/IgG-positive cell ratio was significantly higher in the IgG4-RD group (*p* < 0.001). The IgA-positive cell count and IL-6 expression intensity were higher in the PC-iMCD group than in the IgG4-RD group (*p* < 0.001). Based on these findings, we proposed a new diagnostic approach to differentiate lung lesions of PC-iMCD and IgG4-RD. Our approach can be utilized to stratify patients with suspected lung-dominant PC-iMCD to identify candidates for strong immunosuppressive treatment, including IL-6 blockade, at an early stage.

## 1. Introduction

Castleman disease (CD) was originally described by Castleman et al. in 1956 [1]. It is an umbrella term for heterogeneous lymphoproliferative disorders that share several basic histological features with varying clinical manifestations. Unicentric CD is localized to a single anatomical region of lymph nodes, whereas multicentric CD (MCD) involves multiple lymph node regions and is typically associated with systemic inflammatory symptoms and abnormal laboratory findings [2]. The etiology of MCD remains mostly unknown, except for some cases related to Kaposi’s sarcoma-associated herpesvirus/human herpes virus-8 (HHV-8), and idiopathic cases are known as idiopathic MCD (iMCD). The typical clinical presentation of iMCD includes fever, general fatigue, anemia, elevated C-reactive protein levels (CRP), and hypergammaglobulinemia [3]. Previous studies have shown an association between systemic inflammation and high serum interleukin (IL)-6 levels [4,5,6]. There is a spectrum of histopathological features observed in iMCD that are often separated into hypervascular (HV) type, plasma cell (PC) type, and mixed type [4]. Patients with HV type iMCD exhibit regressed germinal centers and hypervascularization without prominent plasmacytosis called Castleman-Kojima disease (iMCD with thrombocytopenia, ascites, reticulin fibrosis, renal dysfunction, organomegaly (TAFRO) symptoms). By contrast, those with PC-iMCD generally exhibit expanded intrafollicular areas with sheet-like infiltration of plasma cells [4,7]. Although PC-iMCD typically exhibits systemic lymphadenopathy, it occasionally manifests as parenchymal lung disease [8,9].

Immunoglobulin (Ig) G4-related disease (IgG4-RD) is a systemic immune-mediated disorder characterized by elevated serum IgG4 levels and mass-forming lesions [10]. This condition frequently involves multiple organs such as the pancreas, lacrimal and salivary glands, and bile ducts. While relatively rare, lung involvement has been reported in this disease [11,12]. Histological hallmarks of IgG4-RD are lymphoplasmacytic infiltration with abundant IgG4-positive cells and fibrosis [10,13].

iMCD is a heterogeneous disease that exhibits clinicopathological overlap with other conditions, such as autoimmune disorders, infectious diseases, and IgG4-RD [4]. Differentiating between PC-iMCD and IgG4-RD, in particular, is challenging. Clinically, patients with PC-iMCD frequently exhibit elevated serum IgG4 levels. Histologically, lymph node specimens of PC-iMCD often display marked lymphoplasmacytic infiltration accompanied by an increased number of IgG4-positive cells. As a result, they often fit the diagnostic criteria for IgG4-RD [14,15]. Because of these diagnostic issues, clinicians are faced with the problem of differentiating between the two diseases, as these conditions require different treatment strategies. Patients with IgG4-RD generally respond well to corticosteroid monotherapy, whereas those with PC-iMCD usually fail to respond to initial corticosteroid therapy, as is substantiated by a previously reported failure rate of 54% [16]. IL-6 receptor blockade therapy is often required for iMCD cases [16,17,18].

Increasing difficulties are encountered in differentiating lung lesions due to PC-iMCD from those due to IgG4-RD. To date, only two studies have investigated the clinical and histological differences between lung lesions of PC-iMCD and IgG4-RD [19,20]. Histological analyses of lung lesions of PC-iMCD and IgG4-RD remain inconclusive due to the small number of samples used in these studies. To our knowledge, the diagnostic utility of IL-6 and IgA immunostaining to differentiate between lung lesions of PC-iMCD and IgG4-RD has not been adequately investigated. 

Here, we present detailed clinical and histological analyses of the lung lesions associated with PC-iMCD and IgG4-RD, with particular reference to the expression patterns of various markers, including IgA and IL-6, and propose a new diagnostic approach for lung PC-iMCD. 

## 2. Materials and Methods

### 2.1. Case Selection

Twenty-three Japanese patients with lung lesions caused by either PC-iMCD (n = 16) or IgG4-RD (n = 7) were included in this study. All 23 cases were retrieved from surgical pathology consultation files from the Department of Pathology, Okayama University Hospital. All patients underwent surgical lung resection by video-assisted thoracic surgery except for one patient with IgG4-RD who was autopsied.

### 2.2. Diagnostic Criteria

Patient data were retrospectively reviewed to determine if they were compatible with the latest criteria of either PC-iMCD or IgG4-RD [4,10,21]. iMCD was diagnosed based on a combination of laboratory and clinical criteria of the consensus diagnostic criteria for iMCD [4] and the presence of enlarged lymph nodes in ≥2 lymph node stations. Lymph node specimens were obtained in 8 of the 16 patients with PC-iMCD, and all specimens histologically showed sheet-like infiltration of mature plasma cells in the interfollicular area, which is consistent with that in PC-iMCD. Lymph node specimens were not obtained from the other eight patients for anatomical reasons. HHV-8-associated MCD was excluded either serologically or via immunostaining for HHV-8 using an anti-HHV-8 antibody (13B10, 1:40; LifeSpan Biosciences, Seattle, WA, USA). Patients with autoimmune diseases, malignancies, and infectious diseases were clinically excluded from this study. No patients with both diseases had light chain restriction with in situ hybridization.

All patients with IgG4-RD met both the 2019 American College of Rheumatology/European League Against Rheumatism (ACR/EULAR) Classification Criteria for IgG4-Related Disease [21] and the following criteria: serum IgG4 elevation (>135 mg/dL) and histological findings of lymphoplasmacytic infiltration with abundant IgG4-positive cells (>50/high-power fields (HPFs) and IgG4/IgG-positive cell ratio > 40%) [10].

### 2.3. Analysis of Clinical Features and CT Findings

Electronic medical records and consultation files of the study subjects were reviewed for age, sex, symptoms, affected organs, laboratory data (white blood cells (WBCs), hemoglobin (Hb), platelets (Plts), albumin (Alb), C-reactive protein (CRP), Igs (IgG, IgG4, IgA, IgE), and IL-6), as well as a treatment course. Lymph node involvement and the affected organs were clinically determined by physical examination and CT imaging at the time of surgical biopsy. Radiologists evaluated the chest CT findings obtained from each institution. Using imaging reports and radiologists’ remarks, we collected information related to the localization, distribution patterns, presence of ground-glass opacity (GGO), nodular shadows, pleural effusions, and differential diagnoses. 

### 2.4. Histological Evaluation

All lung specimens were fixed with 10% formaldehyde and embedded in paraffin. These paraffin-embedded tissue blocks were sliced into 4-µm-thick sections, which were stained with hematoxylin and eosin and either Elastica-Masson Goldner or Elastica-van Gieson (EVG) stains. 

Lesion distribution patterns (bronchovascular bundles (BvBs), interlobular septal, and alveolar interstitial), fibrosis types, number of lymphoid follicles and eosinophils, and the presence of hemosiderin-laden macrophages, as well as obliterative vasculitis determined by elastic staining with either Elastica-Masson Goldner or EVG staining, were histologically examined. The numbers of lymphoid follicles and obliterative vasculitis per slide were determined. The number of eosinophils in areas with the highest density of eosinophils was counted in three different HPFs (eyepiece, 10× and lens, 40×), and average values were calculated. 

### 2.5. Immunohistochemistry

Immunohistochemistry was performed using an automated Bond-III instrument (Leica Biosystems, Wetzlar, Germany) with the following primary antibodies: IL-6 (10C12, 1:200; Novo castra, Newcastle, UK); IgG4 (HP6025, 1:10,000; The Binding Site, Birmingham, UK); IgG (RWP49, 1:600; Novo castra, Newcastle, UK); polyclonal IgA (polyclonal, 1:20,000; Dako, Glostrup, Denmark).

The number of IgG4-positive cells and IgG-positive cells was estimated in areas with the highest density of IgG4-positive cells. As stipulated in the consensus statement of the pathology of IgG4-RD published in 2012 [10], three different HPFs were examined, and the average was calculated to determine the number of IgG4-positive cells and the IgG4/IgG-positive cell ratio. The number of IgA-positive cells in areas with the highest density of IgA-positive cells was counted in three different HPFs, and the averages were calculated. IL-6 staining intensities in both the interfollicular and germinal center area were scored as negative “0”, dim positive “1”, positive “2”, and strongly positive “3” by two independent pathologists until a consensus was reached.

### 2.6. Statistical Analysis

Statistical analyses were conducted using the Mann–Whitney U test or Fisher’s exact test. Statistical significance was set at *p* < 0.05. All statistical analyses were performed using SPSS software, version 24 (SPSS, Chicago, IL, USA).

### 2.7. Ethical Approval

The study protocol was approved by the Institutional Review Board of Okayama University, Okayama, Japan (IRB approval number: 1607-016, 2007-033). The study wasperformed in accordance with the ethical standards laid down in the Declaration of Helsinki. Informed consent was obtained for all subjects in the form of opt-out on the website. 

## 3. Results

### 3.1. Clinical and Laboratory Findings

The demographics, clinical and laboratory findings of the patients are summarized in Table 1.

Among patients with PC-iMCD, there were seven men and nine women aged 26–73 years with a median age of 49.5 years. All patients with IgG4-RD were men aged 39–77 years with a median age of 70.0 years. Patients with PC-iMCD were significantly younger than those with IgG4-RD (*p* = 0.018). With respect to comorbidities, patients with PC-iMCD had IgA nephropathy related to CD (2/16; 12.5%), asthma (2/16; 12.5%), emphysema (1/16; 6.3%), chronic hepatitis B (1/16; 6.3%), type 2 diabetes mellitus (1/16; 6.3%), and hyperthyroidism (1/16; 6.3%). Patients with IgG4-RD had asthma (4/7; 57.1%), type 2 diabetes mellitus (3/7; 42.9%), allergic rhinitis (2/7; 28.6%), hypertension (1/7; 14.3%), chronic kidney disease (1/7; 14.3%), previous history of lung cancer (1/7; 14.3%), and old myocardial infarction (1/7; 14.3%). Six of the seven (85.7%) patients with IgG4-RD suffered from allergic diseases, including asthma or allergic rhinitis, whereas two of the sixteen (12.5%) patients with PC-iMCD had allergic diseases (*p* = 0.002). Seven of the sixteen patients with PC-iMCD (43.8%) had constitutional symptoms such as general fatigue or fever, whereas no patients with IgG4-RD presented with these symptoms (*p* = 0.057). 

In the PC-iMCD group, the involvement of extranodal organs including the skin, spleen, and liver were observed. By contrast, no patients in the IgG4-RD group had skin rash or hepatosplenomegaly. Instead, six of the seven (85.7%) patients with IgG4-RD had typical extrapulmonary lesions in at least one of the following: the salivary glands, lacrimal glands, pancreas, bile duct, and aorta. Particularly, the pancreas and salivary glands were more frequently affected in patients with IgG4-RD than those with PC-iMCD (*p* = 0.02 and *p* = 0.003, respectively). 

All the patients with PC-iMCD showed lymph node enlargement (short-axis diameter > 1 cm) in ≥2 lymph node stations in the chest CT images. In four (25.0%) patients, the extent of lymph node enlargement was limited to the hilar and mediastinal regions. Eight patients (50%) showed evidence of PC-iMCD in the lymph nodes. In these patients, the following lymph nodes were biopsied: mediastinal (five patients), inguinal (two patients), and axillary (one patient). Notably, in one patient with PC-iMCD, lung lesions preceded cervical, mediastinal, and hilar lymph node swelling. Another patient initially exhibited only lung lesions and submandibular lymphadenopathy, and inguinal lymphadenopathy resulted over time. No patients in the IgG4-RD group exhibited systemic lymph node enlargement, while five (71.4%) showed localized lymph node swelling. The affected lymph nodes were limited to the submandibular, mediastinal, or hilar lymph nodes. One patient with IgG4-RD had preceding lung lesions. The patient presented with the submandibular, pancreas, and periaortic lesions approximately two years after the emergence of lung lesions and surgical lung biopsy. 

Laboratory findings indicated that the levels of Plts, CRP, serum IgG, and serum IgA were significantly elevated in the PC-iMCD group compared to those in the IgG4-RD group (*p* < 0.001, *p* < 0.001, *p* = 0.033, *p* < 0.001, respectively). By contrast, albumin, serum IgG4, and the serum IgG4/IgG ratio were significantly higher in the IgG4-RD group (*p* = 0.019, *p* = 0.018, *p* < 0.001, respectively). Importantly, all patients with PC-iMCD for whom serum IgG4 levels were available (15 cases) had elevated serum IgG4 level (>135 mg/dL). There were no significant differences in serum IgE levels between the two groups (*p* = 1.000). The serum IL-6 levels were elevated (>4 pg/mL) in all patients with PC-iMCD, whereas IL-6 data were not available for evaluation in the IgG4-RD group.

### 3.2. Evaluation of Chest CT Images

The CT findings of PC-iMCD and IgG4-RD groups are shown in Appendix A. All the patients except for one with PC-iMCD displayed diffuse bilateral lung lesions. One patient with PC-iMCD had a localized 1.5-cm mass lesion in the right upper lobe. The CT findings were mostly a mixture of GGO and nodular shadows (mostly small nodular lesions < 1 cm or granular lesions). Most patients in both groups (15/16 (93.8%) and 5/7 (71.4%) for PC-iMCD and IgG4-RD, respectively) had lymphatic tract lesions, including thickening of BvBs, interlobular septa, and pleura. Thus, differential diagnoses based on CT were very diverse and included lymphoproliferative disorders, lymphoma, sarcoidosis, collagenous diseases, and pulmonary edema. 

### 3.3. Analysis of Histological Features

Representative pictures of lung lesions in patients with PC-iMCD and IgG4-RD are shown in Figure 1, and a comparison of histological features in both groups is summarized in Table 2. 

In both groups, histological features of lung lesions were associated with plasma cell and lymphocyte infiltration. The sheet-like proliferation of mature plasma cells was more frequently confirmed in patients with PC-iMCD than those with IgG4-RD (*p* = 0.027). In patients with IgG4-RD, a mixed infiltrate of immature (plasmablast-like cells with dispersed chromatin and a single prominent nucleolus) to mature plasma cells, small lymphocytes, and eosinophils were observed. Disease distribution patterns in both groups mainly exhibited lymphangitic spread, including BvBs, interlobular septa, and pleura. Additionally, 7/16 (43.8%) patients with PC-iMCD and 3/7 (42.9%) patients with IgG4-RD displayed alveolar interstitial involvement. Nodular patterns were seen in areas of marked inflammation. There was no significant difference in the number of lymphoid follicles between the two groups (*p* = 0.053). Patients in both groups exhibited fibrosis to varying degrees, and there was no difference in the types of fibrosis between the two groups. Typical storiform fibrosis was seen only in one patient with PC-iMCD (Appendix A). Obliterative vasculitis, which is considered as a pathognomonic change associated with IgG4-RD, was commonly observed in both groups (12/16 (75%) and 4/7 (57.1%) for patients with PC-iMCD and IgG4-RD, respectively). There was no significant difference between the number of obliterative vasculitis per unit square in the two groups (*p* = 0.892). Only two patients with PC-iMCD exhibited the presence of hemosiderin-laden macrophages, while none were observed in the IgG4-RD group did. Eosinophil infiltration was significantly greater in the IgG4-RD group (*p* = 0.004). Based on the number of eosinophils/HPF, a receiver operating characteristic (ROC) curve differentiating PC-iMCD from IgG4-RD was generated. The area under the curve (AUC) was 0.875. The curve had a cut-off value of 19.5/HPF, as determined using the Youden index (sensitivity + specificity − 1); this value was associated with a sensitivity of 75.0% and a specificity of 100%.

### 3.4. Analysis of Immunohistochemical Features

A comparison of immunohistochemical findings is shown in Figure 2. The number of IgG4-positive cells in both groups was found to be increased, and there was no significant difference between the two groups (*p* = 0.867). The IgG4/IgG-positive cell ratio was significantly higher in the IgG4-RD group (*p* < 0.001). In the IgG4-RD groups, the ratio was approximately 100% in all patients, with the highest percentage of 124%. 

The number of IgA-positive cells was significantly higher in the PC-iMCD group than that in the IgG4-RD group (*p* < 0.001) (Figure 2 and Appendix A).

Based on the IgG4/IgG-positive cell ratio (%) and the number of IgA-positive cells/HPF, ROC curves, which differentiated PC-iMCD from IgG4-RD, were generated. With respect to the IgG4/IgG-positive cell ratio, the AUC was estimated to be 0.982, and the curve had a cut-off value of 89.2%; the value was associated with a sensitivity and specificity of 93.8% and 100%, respectively. The AUC for the number of IgA-positive cells/HPF was 0.982, the-cut off value was 23.7/HPF, and this value was associated with 97.5% sensitivity and 100% specificity.

IL-6 expression via immunohistochemistry in both groups is shown in Figure 3. Fourteen of 16 (87.5%) PC-iMCD patients had a score of 2 to 3, with strong, positive granular IL-6 expression in the interfollicular plasma cell cytoplasm. In the IgG4-RD patients, IL-6 immunostaining was negative except for two patients who showed weak, positive IL-6 immunoreactivity in some plasma cells in the interfollicular area. IL-6 expression intensity scores in the interfollicular and the germinal center areas were significantly higher in the PC-iMCD group than those in the IgG4-RD group (*p* < 0.001, *p* = 0.002, respectively; Figure 4).

### 3.5. Analysis of Treatment and Clinical Course

Clinical courses for the patients, except two patients in the PC-iMCD group for whom the data were unavailable, are summarized in Table 3. Although 11 of the 14 (78.6%) patients with PC-iMCD were initially treated with corticosteroid, none of them showed a complete clinical response to corticosteroid monotherapy. Five patients were treated with tocilizumab after initial corticosteroid monotherapy, and all five patients exhibited rapid clinical remission; CT findings of these patients also improved. Two patients (Cases 12 and 13 of PC-iMCD) were not administered corticosteroid therapy because they refused treatment or had minor symptoms. Histological findings of one PC-iMCD patient (Case 2) are shown in Appendix A. The patient exhibited elevated serum IgG4 level (1640 mg/dl) and an increased number of IgG4-positive cells in the tissue (the average of three different HPFs; 205/HPF), while exhibiting an increased number of IgA-positive cells (the average of three different HPFs; 78.7/HPF) as well as positive IL-6 expression. The patient did not respond to initial corticosteroid monotherapy, but the use of tocilizumab led to rapid clinical remission. Six of seven patients in the IgG4-RD group were treated with corticosteroids, and all exhibited rapid remission. One patient (Case 7 of IgG4-RD), who was diagnosed with IgG4-RD one month before death, died of non-occlusive mesenteric ischemia due to IgG4-RD (an autopsy revealed diffuse IgG4-RD-related obliterative vasculitis in the mesentery as well as plasma cell/eosinophil infiltrates in the mesenteric arteries) before treatment could be administered.

## 4. Discussion

Diagnosing lung-dominant PC-iMCD is challenging for both clinicians and pathologists. From a clinical perspective, because of the non-specificity of radiological findings of iMCD [22], imaging studies do not sufficiently rule out other diseases including IgG4-RD and malignancies [23,24,25]. Although lymph node findings are considered necessary for pathologists to diagnose PC-iMCD, diagnostic lymph node excision may not always be feasible in some cases. In these cases, the results of a clinicopathological analysis of lung lesions may increase clinical suspicion for PC-iMCD and therefore may be used to influence treatment decisions.

All 16 patients with PC-iMCD in our study showed enlarged lymph nodes in at least two lymph node stations. However, 8 of 16 (50%) patients did not receive a histological diagnosis based on lymph node specimens because of inaccessibility caused by anatomical factors, as described above. Although these eight patients did not fully meet the consensus criteria for iMCD [4] without lymph node findings, they showed a similar clinical course and histological findings such as lung lesions common to PC-iMCD. Therefore, these patients could also be considered to have PC-iMCD, and lung-specific complementary criteria should be established to fill the current diagnostic gap.

IgG4-RD is the most challenging differential diagnosis of PC-iMCD because of the similarities in the histological features and lack of specific biomarkers. As both PC-iMCD and IgG4-RD are rare, only two studies have previously investigated the clinicopathological differences between the lung lesions in these two diseases [19,20]. One study involved lung specimens from 15 patients with iMCD (14 surgically resected and 1 autopsied) and 9 with IgG4-RD (all surgically resected) [19], while the other included lung biopsy specimens from 8 patients with iMCD (7 video-associated thoracoscopic biopsies and 1 transbronchial biopsy) and 16 with IgG4-RD (10 transbronchial biopsies, 5 video-associated thoracoscopic biopsies, and 1 percutaneous needle biopsy) [20]. Because of the small number of cases in these studies, the histological features of lung lesions of iMCD and IgG4-RD remain controversial.

In this study, we compared the clinicopathological findings associated with the lung lesions of PC-iMCD and IgG4-RD. To our knowledge, this is the first study to immunohistochemically assess the expression patterns of IL-6 and IgA in the lung lesions of both diseases, specifically to improve diagnostic utility.

To develop comprehensive clinical perspectives, we detailed the patient demographics and laboratory findings and summarized the findings. Clinically, the median value for age; involvement of the pancreas or salivary gland; serum levels of Plts, Alb, CRP, IgG, and IgG4; and IgG4/IgG ratio, significantly differed between the two groups, as previously reported [14,15,26], supporting the validity of our study design and population.

Histologically, there were significant differences in the presence of sheet-like plasmacytosis and the number of eosinophils between the patients of PC-iMCD and IgG4-RD (*p* = 0.027 and *p* = 0.004, respectively). Although a previous study suggested that the degree of active fibrosis and the number of obliterative vasculitis were significantly different between iMCD and IgG4-RD groups [19], our study revealed no significant differences between the groups with regard to these features (*p* = 0.124 and *p* = 0.892, respectively). This may be because we comprehensively evaluated entire sections, whereas the previous study evaluated values in three randomly selected visual fields. Furthermore, although one study suggested that hemosiderin-laden lesions in lymph node specimens can be used to distinguish iMCD from IgG4-RD [27], this finding was confined to only two patients (12.5%) with PC-iMCD. As the diagnostic value of these findings in lung lesions is uncertain, pathologists should not rely on only single measures such as obliterative vasculitis and the fibrosis type.

Immunohistochemically, the following three measurements are considered particularly important: (1) the IgG4/IgG-positive cell ratio; (2) IgA-positive cells; (3) IL-6 expression intensity. The IgG4/IgG-positive cell ratio was significantly higher in the IgG4-RD group than in the PC-iMCD group (*p* < 0.001). The number of IgA-positive cells was higher in the PC-iMCD group (*p* < 0.001). A previous study suggested that IgA immunohistochemistry findings can be used to distinguish lymph node lesions of PC-iMCD from those of IgG4-RD [28]. 

In IL-6 immunostaining, the intensity scores for interfollicular plasma cells and cells of the germinal center area were significantly higher in the PC-iMCD group than in the IgG4-RD group (*p* < 0.001 and *p* = 0.002, respectively). One study that performed immunostaining on lymph node specimens reported that plasma cells in PC-type CD cases showed significantly increased IL-6 expression compared to those in reactive lymph nodes [29], supporting our result. Overall, the extent of IL-6 expression in interfollicular plasma cells may be a useful diagnostic indicator of PC-iMCD. 

Based on our clinicopathological findings, we propose a new differential diagnostic approach for PC-iMCD compared with IgG4-RD in the lungs (Table 4). With respect to clinical findings, constitutional symptoms such as fever or general fatigue are suggestive of PC-iMCD, whereas the presence of salivary gland or pancreatic lesions suggest IgG4-RD. Elevated CRP and serum IgA levels (above the normal range), which are characteristic findings of hyper IL-6 syndromes including iMCD, have been reported to be unlikely in patients with IgG4-RD [30]. In our study, serum IgA levels were elevated over the normal range (>410 mg/dL) in 11 out of 15 (73.3%) patients with PC-iMCD for whom serum IgA levels were available, but in none of those with IgG4-RD included in our study. Cut-off values determined using histological and immunohistochemical findings were adjusted to an approximate integer for ease of use in clinical practice. Obliterative vasculitis and active fibrosis, considered as characteristics of IgG4-RD, did not contribute to the diagnosis of PC-iMCD and IgG4-RD via lung lesions. As histological findings suggestive for PC-iMCD may also be common to other hyper IL-6 syndromes, such as rheumatoid arthritis, Sjögren syndrome, and systemic lupus erythematosus, these diseases should be clinically excluded before applying our approach. Moreover, it needs to be noted that the latest criteria for iMCD [4] and IgG4-RD [21] remain to be the cornerstones to discriminate these two entities in general.

Our study had several limitations. First, this was a retrospective study with relatively small sample size. Because of the rarity of lung involvement in PC-iMCD and IgG4-RD, selection bias may be involved. Second, IL-6 immunostaining patterns in other hyper IL-6 syndromes were not investigated in this study. As these syndromes also involve abundant IgG4-positive cells [31], such diseases should be carefully ruled out. Despite these limitations, our study provides novel insights into issues associated with diagnosing lung lesions in PC-iMCD separately from those in IgG4-RD.

In conclusion, we investigated the detailed clinicopathological features of lung lesions in PC-iMCD and compared the findings with those in IgG4-RD. The usefulness of immunostaining was highlighted, with specific reference to IL-6, IgA, and the IgG4/IgG-positive cell ratio. The differential diagnostic approach proposed in this study may be useful for accurately distinguishing lung lesions of PC-iMCD from those of IgG4-RD at an early stage, particularly in lung-dominant PC-iMCD where excisional lymph node biopsy is not feasible. Future studies using a larger patient sample size are needed to investigate the clinicopathological findings of lung lesions in other PC-iMCD mimickers, including hyper-IL-6 syndromes, to establish lung-specific criteria for PC-iMCD. 

## Figures and Tables

**Figure 1 jpm-10-00269-f001:**
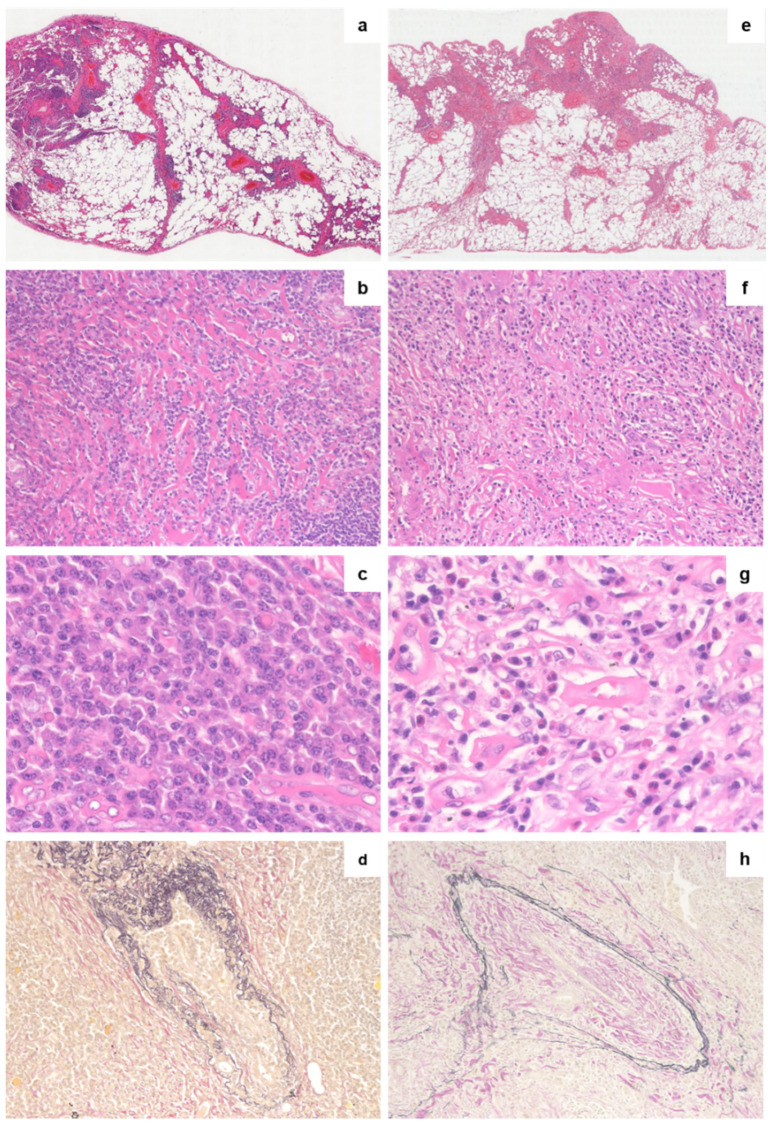
Comparison of histopathological features of lung lesions in PC-iMCD and IgG4-RD. Representative pictures of lung lesions in PC-iMCD (**a**–**d**) and IgG4-RD (**e**–**h**) groups are shown. (**a**,**e**) HE stains showing a lymphangitic spread pattern (low-power view). (**b**) Hyalinized fibrosis (HE, 200×). (**c**) The sheet-like proliferation of mature plasma cells in the interfollicular area (HE, 400×). (**d**,**h**) Obliterative vasculitis (EVG, 400×); **f**, active fibrosis (HE, 200×). (**g**) Lymphoplasmacytic infiltration with the increased number of eosinophils (HE, 400×). HE, hematoxylin and eosin; EVG, Elastica-van Gieson.

**Figure 2 jpm-10-00269-f002:**
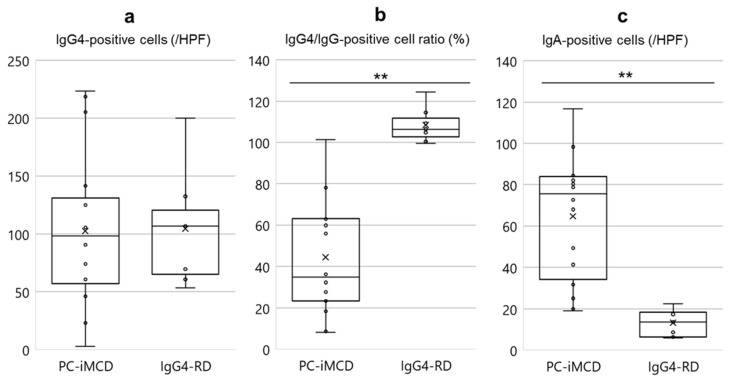
Comparison of immunohistochemical findings of PC-iMCD and IgG4-RD. (**a**) There was no significant difference in the number of IgG4-positive cells (*p* = 0.867). (**b**) The IgG4/IgG-positive cell ratio was higher in the IgG4-RD group (*p* < 0.001). (**c**) The number of IgA-positive cells was more significant in the PC-iMCD group (*p* < 0.001). The number of IgG4- and IgG-positive cells were estimated in areas with the highest density of IgG4-positive cells. The number of IgA-positive cells was estimated in areas with the highest IgA-positive cell density. The average of three different HPFs was used to obtain the number of IgG4- and IgA- positive cells (/HPF) and IgG4/IgG-positive cell ratio (%). Data are shown in boxplots with whiskers representing minima to maxima. Differences between the two groups were evaluated by the Mann‒Whitney U test (** *p* < 0.001). HPF, high-power field.

**Figure 3 jpm-10-00269-f003:**
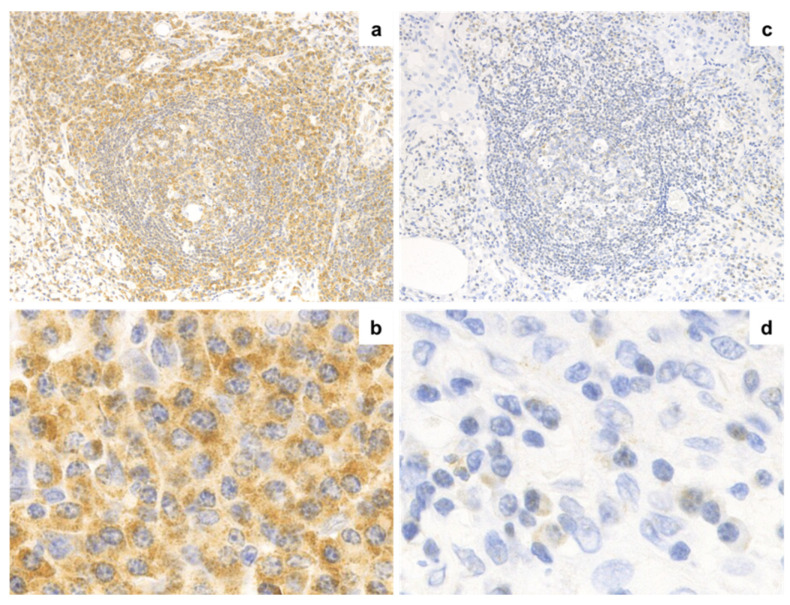
Immunohistochemical analysis of IL-6 expression. IL-6 expression via immunohistochemistry in PC-iMCD ((**a**), 200× and (**b**), 400×) and IgG4-RD ((**c**), 200× and (**d**), 400×) groups is shown. (**a**,**b**) IL-6 was strongly expressed in most cells in the interfollicular and germinal center area (score 3). Positive cells showed granular cytoplasmic staining. (**c**,**d**) Certain cells in the interfollicular and germinal center area were “dim positive” for IL-6 (Score 1). IL-6, Interleukin-6; HPF, high-power field.

**Figure 4 jpm-10-00269-f004:**
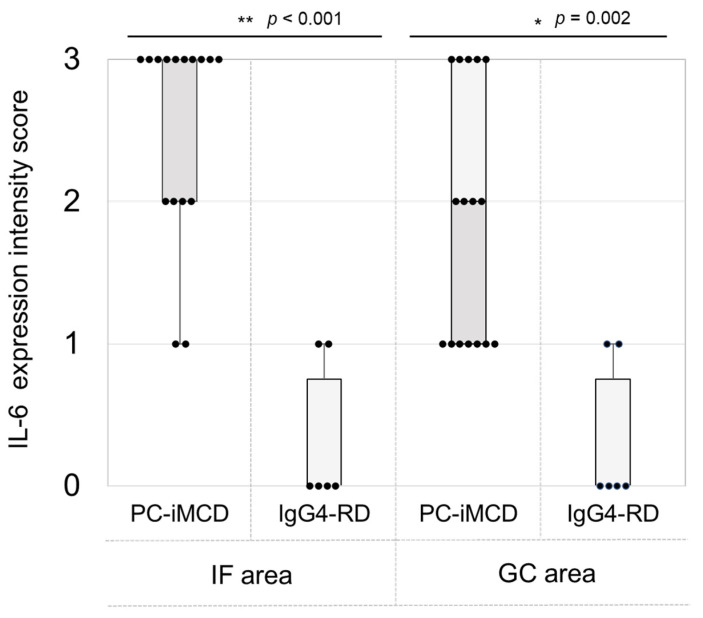
Comparison of IL-6 expression intensities. IL-6 immunostaining was available for 16 PC-iMCD and 6 IgG4-RD cases. The intensity of IL-6 staining in interfollicular (IF) and germinal center (GC) areas was assessed by two pathologists. Key: 0 = negative, 1 = dim positive, 2 = positive, 3 = strongly positive. IL-6 intensity scores in the IF and the GC areas were significantly higher in the PC-iMCD group than those in the IgG4-RD group (*p* < 0.001, *p* = 0.002, respectively). The graph depicts individual values and boxplots with whiskers representing minima to maxima. Differences between the two groups were evaluated by the Mann-Whitney U test (* *p* < 0.05, ** *p* < 0.001).

**Table 1 jpm-10-00269-t001:** Comparison of clinical findings.

	PC-iMCD (*n* = 16)	IgG4-RD (*n* = 7)	*p-*Value	
Age (median ± SD)	49.5 ± 12.0	70.0 ± 12.5	**0.018 ***	
Sex (M/F)	7/9	7/0	―	
Symptom, *n* (%)				
Cough	6 (37.5)	1 (14.3)	0.366	
General fatigue	5 (31.3)	0	0.272	
Fever	4 (25.0)	0	0.273	
Shortness of breath	3 (18.8)	2 (28.6)	0.621	
Skin rash	3 (18.8)	0	0.526	
Hemoptysis	1 (6.3)	0	1.000	
Affected organs, *n* (%)				
Lung	16 (100)	7 (100)	―	
Lymph nodes	16 (100)	5 (71.4)	0.083	
Skin	3 (18.8)	0	0.526	
Spleen	2 (12.5)	0	1.000	
Liver	2 (12.5)	0	1.000	
Ureter	1 (6.3)	1 (14.3)	0.526	
Retroperitoneum	0	1 (14.3)	0.304	
Kidney	0	2 (28.6)	0.083	
Aorta(periaortitis)	0	2 (28.6)	0.083	
Bile duct	0	2 (28.6)	0.083	
Lacriminal gland	0	2 (28.6)	0.083	
Pancreas	0	3 (42.9)	**0.020 ***	
Salivary gland	1 (6.3)	5 (71.4)	**0.003 ***	
Laboratory findings (median ± SD)				
WBC (/μL)	7800 ± 2059	7600 ± 1591	0.671	
Hb (g/dL)				
Male	13.0 ± 2.1	14.1 ± 1.7	0.275	
Female	11.1 ± 1.2	―	―	
Plt (×10^4^/µL)	39.2 ± 9.6	21.4 ± 4.5	**<0.001 ****	
Alb (g/dL)	3.0 ± 0.5	3.8 ± 0.7	**0.019 ***	
CRP (mg/dL)	3.8 ± 3.7	0.3 ± 0.6	**<0.001 ****	
IgG (mg/dL)	3883 ± 1242	2810 ± 1192	**0.033 ***	
IgG4 (mg/dL)	374 ± 479 ^†^	1050 ± 590	**0.018 ***	
IgG4/IgG (%)	11.3 ± 9.4	37.4 ± 12.8	**<0.001 ****	
IgA (mg/dL)	567 ± 331	170 ± 74.8 ^‡^	**<0.001 ****	
IgE (IU/mL)	1082 ± 870 ^†^	771 ± 1800 ^‡^	1.000	
IL-6 (pg/mL)	10.1 ± 8.5 ^†^	ND	―	

^†^ IgG4, IgE, and IL-6 levels were available for 15, 14, and 12 patients with Plasma cell type idiopathic multicentric Castleman disease (PC-iMCD), respectively. ^‡^ IgA and IgE were available for six patients with immunoglobulin G4-related disease (IgG4-RD). Significance was calculated using the Mann‒Whitney U test. Fisher’s exact analysis was used for the statistical analysis of nominal scales. * *p* < 0.05, ** *p* < 0.001. Significant *p*-values are in bold. SD, standard deviation; WBC, white blood cell; Hb, hemoglobin; Plt, platelet; Alb, albumin; CRP, C-reactive protein; Ig, immunoglobulin; IL-6, interleukin 6; ND, not done. Normal ranges: WBC, 3900–9800/µL; Hb, 13.4–17.6 g/dL (male), 11.3–15.2 g/dL (female); Plt, 12.7–35.6 × 10^4^/µL; Alb, 3.7–5.2 g/dL; CRP, 0.00–0.30 mg/dL; IgG, 870–1700 mg/dL; IgG4, 4.8–105 mg/dL; IgA, 110–410 mg/dL; IgE, 0–170 IU/mL; IL-6, 0–4.0 pg/mL.

**Table 2 jpm-10-00269-t002:** Comparison of histological findings.

	PC-iMCD (*n* = 16)	IgG4-RD (*n* = 7)	*p*-Value
Distribution pattern, *n* (%)			
BvBs	16 (100)	7 (100)	NS
Interlobular septal	15 (93.8)	5 (71.4)	0.209
Alveolar interstitial	7 (43.8)	3 (42.9)	1.000
Nodular	2 (12.5)	1 (14.3)	1.000
Lymphoid follicles(/cm^2^) (median ± SD)	74.0 ± 75.2	33.3 ± 37.2	0.053
Sheet-like plasmacytosis, *n* (%)	11 (68.8)	1 (14.3)	**0.027 ***
Fibrosis			
Active fibrosis, *n* (%)	10 (62.5)	7 (100)	0.124
Dense hyalinized fibrosis, *n* (%)	11 (68.8)	3 (42.9)	0.363
Hemosiderin laden macrophages, *n* (%)	2 (12.5)	0 (0)	1.000
Obliterative vasculitis (/cm^2^) (median ± SD)	13.3 ±16.1	20.8 ± 17.7	0.892
Eosinophils (/HPF) (median ± SD) ^†^	10.0 ± 19.0	38.7 ± 24.2	**0.004 ***

^†^ The number of eosinophils (/HPF) is the average of three different HPFs with the highest density. Significance was calculated using the Mann‒Whitney U test. Fisher’s exact test was used for the analysis of nominal scales. * *p* < 0.05. Significant *p*-values are in bold. BvB, bronchovascular bundle; HPF, high-power field; NS, no significant difference.

**Table 3 jpm-10-00269-t003:** Treatment and clinical courses of PC-iMCD and IgG4-RD patients.

Case Number	Age/Sex	Initial Therapy	Response to PSL	Additional Treatment	Outcome ^†^	Observation Period (mo)
PC-iMCD						
1	46/M	PSL 10 mg	No response	Tocilizumab	remission	83
2	40/F	PSL 40 mg	No response	Tocilizumab	remission	93
3	53/M	PSL 30 mg	No response	Tocilizumab	remission	90
4	44/M	PSL 30 mg	Repeatedly worsened during tapering	Tocilizumab	remission	90
5	48/M	PSL 30 mg	No response	Tocilizumab	remission	16
6	64/F	PSL 30 mg	No response	Tacrolimus	progression	18
7	37/F	PSL 60 mg	No response	No additional treatment	progression	46
8	41/F	PSL 20 mg	No response	No additional treatment	no change	12
9	43/F	PSL 25 mg	Repeatedly worsened during tapering	No additional treatment	no change	38
10	54/F	PSL 40 mg	Partial improvement	No additional treatment	partial remission	180
11	60/M	PSL 40 mg	Partial improvement	No additional treatment	partial remission	13
12	52/M	No treatment	―	―	progression	88
13	51/M	No treatment	―	―	progression	48
14	73/F	No treatment	―	―	remission	36
IgG4-RD						
1	39/M	PSL 40 mg	Improvement	No additional treatment	remission	61
2	73/M	PSL 30 mg	Improvement	No additional treatment	remission	88
3	70/M	PSL 40 mg	Improvement	No additional treatment	remission	132
4	64/M	PSL 20 mg	Improvement	No additional treatment	remission	132
5	71/M	PSL 20 mg	Improvement	No additional treatment	remission	52
6	67/M	PSL 30 mg	Improvement	No additional treatment	remission	31
7	77/M	No treatment	―	No additional treatment	Dead	2

Treatment information was available for 14 PC-iMCD patients and 7 IgG4-RD patients. ^†^ “Outcome” indicates the clinical status of patients as of their last visit to the hospital; “Remission” represents the status wherein laboratory findings, subjective symptoms, and radiographic findings of patients were all improved; “Partial remission” represents the status where laboratory findings did not show sufficient improvement to reach the normal range, and the subjective symptoms or radiographic findings of patients remained unchanged or worsened; “No change” represents the status where all clinical findings remained unchanged; “Progression” represents the status where all the following findings worsened: laboratory findings, subjective symptoms, or radiographic findings. PSL, prednisolone; mo, months.

**Table 4 jpm-10-00269-t004:** Differential diagnostic approach for lung PC-iMCD in comparison with IgG4-RD.

	Suggestive of PC-iMCD	Suggestive of IgG4-RD
Clinical Findings	Presence of fever or general fatigue	Presence of lesions in salivary gland or pancreas
	CRP elevation of uncertain cause	
	Serum IgA elevation (above normal range)	
Histological Findings	Sheet-like proliferation pattern of mature plasma cells	Eosinophils more than 20/HPF ^†^
Immunohistochemical Findings	Strong expression of IL-6 immunostaining in interfollicular plasma cells and cells in germinal centers IgA-positive cells of more than 24/HPF ^†^	Extremely high IgG4/IgG-positive cell ratio (>90%) ^†^

^†^ Values are the average of three different HPFs. CRP, C-reactive protein; IL-6, interleukin-6; HPF, high-power field.

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
