# Peer review of "Pulmonary Manifestations of Plasma Cell Type Idiopathic Multicentric Castleman Disease: A Clinicopathological Study in Comparison with IgG4-Related Disease"

_jpm, 2020, doi:10.3390/jpm10040269_

Round 1

Reviewer 1 Report

This paper aims to study the differences in pathology between lung manifestations of idiopathic multicentric Castleman disease and IgG4 related disease.  A few comments

  1. The authors use the latest criteria for diagnosis of idiopathic multicentric Castleman disease but do do not use the latest criteria for IgG4RD - The 2019 American College of Rheumatology/European League Against Rheumatism Classification Criteria for IgG4-Related Disease By Wallace et al in 2020. – see in methods and also line 313
  2. Line 229 – how do the authors define immature plasma cells. I am unfamiliar with this terminology.
  3. Line 320 – how was the etiology of the non-occlusive mesenteric ischaemia attributed to IgG4-RD? this is certainly not a typical course of IgG4RD to cause vascular obstruction.
  4. The authors do highlight 4 clinical features and five histological / IHC features that distinguish the 2 diseases. However, since there is such overlap in the pathology, not only in the lung but in other organs such as lymph nodes, the whole panel of diagnostic inclusion and exclusion criteria for IgG4RD (Wallace et al) and criteria for iMCD have to be the final arbiter between the two diseases.  

Author Response

Comment from Reviewer 1

Comment 1: [The authors use the latest criteria for diagnosis of idiopathic multicentric Castleman disease but do do not use the latest criteria for IgG4RD - The 2019 American College of Rheumatology/European League Against Rheumatism Classification Criteria for IgG4-Related Disease By Wallace et al in 2020. – see in methods and also line 313.]

Response 1: Thank you for reviewing our manuscript. We applied the 2019 ACR/EULAR classification criteria for IgG4-RD to our patients and confirmed that all the IgG4-RD patients met the criteria. Regarding line 313, we removed the sentence as there were cases of Castleman disease that did not satisfy the exclusion criteria proposed in the 2019 ACR/EULAR classification criteria for IgG4-RD. We also revised the manuscript accordingly.

Comment 2: [Line 229 – how do the authors define immature plasma cells. I am unfamiliar with this terminology.]

Response2: Thank you for your comment. We defined immature plasma cells as plasmablast-like cells with dispersed chromatin and a single prominent nucleolus morphologically (Int J Rheumatol; 2012:572539.). We have added the definition in the main text.

Comment 3: [Line 320 – how was the etiology of the non-occlusive mesenteric ischaemia attributed to IgG4-RD? this is certainly not a typical course of IgG4RD to cause vascular obstruction.]

Response 3: The case had a definitive diagnosis of IgG4-RD and was planned to be started on corticosteroid. However, before he got the corticosteroid treatment, he complained of severe diffuse abdominal pain and went to the emergency department. He was diagnosed with non-occlusive mesenteric ischemia (NOMI) with abdominal contrast CT and had emergent surgery; he passed away in a few hours despite the intervention. An autopsy was performed, which revealed transmural necrosis of jejunum, ileum, transverse colon, descending colon and  sigmoid colon without apparent obstruction or thrombus formation in arteries. There was diffuse IgG4-RD-related obliterative vasculitis in the mesentery. In the mesenteric arteries, plasma cell/eosinophil infiltrates, the finding often seen in IgG4-RD, were noted in outer membranes. There was no fibrinoid necrosis, neutrophil infiltrates, or granuloma formations in vascular walls, and the patient had negative MPO/PR3-ANCA. While the clinical course is atypical of IgG4-RD, given the autopsy report, we assume that the patient had underlying impaired circulation due to IgG4-RD and some superimposing events which contributed to NOMI.

Comment 4: [The authors do highlight 4 clinical features and five histological / IHC features that distinguish the 2 diseases. However, since there is such overlap in the pathology, not only in the lung but in other organs such as lymph nodes, the whole panel of diagnostic inclusion and exclusion criteria for IgG4RD (Wallace et al) and criteria for iMCD have to be the final arbiter between the two diseases. ]

Response 4: Thank you for pointing this out. In the main text, we have highlighted that the criteria for IgG4-RD by Wallace et al., and the criteria for iMCD are the cornerstones to discriminate these two diseases in general.

Reviewer 2 Report

The manuscript “Pulmonary manifestations of plasma cell type idiopathic 2 multicentric Castleman disease: a clinicopathological study 3 in comparison with IgG4-related disease” by Midori Filiz et al. reports a very useful comparative pathological investigation between these two diseases which may be difficult to distinguish by pathologists.  The manuscript is really well written and illustrated. It clearly gives useful clues for pathologists for diagnosis of these two diseases which have a confusing pattern but need very distinct treatments. One may consider this series of 25 patients too short but these diseases are quite rare and the conclusions by the authors based on pathology, clinical data and response to specific therapies. Therefore I consider this manuscript as adding valuable data and I have really no reservations considering its publication.  

Author Response

Comment from Reviewer 2

Comment 1: [The manuscript “Pulmonary manifestations of plasma cell type idiopathic 2 multicentric Castleman disease: a clinicopathological study 3 in comparison with IgG4-related disease” by Midori Filiz et al. reports a very useful comparative pathological investigation between these two diseases which may be difficult to distinguish by pathologists. The manuscript is really well written and illustrated. It clearly gives useful clues for pathologists for diagnosis of these two diseases which have a confusing pattern but need very distinct treatments. One may consider this series of 25 patients too short but these diseases are quite rare and the conclusions by the authors based on pathology, clinical data and response to specific therapies. Therefore I consider this manuscript as adding valuable data and I have really no reservations considering its publication. ]

Response 1: Thank you for your precious time to review our manuscript. We have revised the manuscript according to the suggestions made by reviewer 1. We hope that it is now acceptable for publication.